# Endophytic Strain *Bacillus subtilis* 26D Increases Levels of Phytohormones and Repairs Growth of Potato Plants after Colorado Potato Beetle Damage

**DOI:** 10.3390/plants10050923

**Published:** 2021-05-05

**Authors:** Antonina Sorokan, Svetlana Veselova, Galina Benkovskaya, Igor Maksimov

**Affiliations:** Institute of Biochemistry and Genetics of the Ufa Federal Research Centre, Russian Academy of Sciences, 71 Oktyabrya avenue, Ufa 450054, Russia; fourtyanns@googlemail.com (A.S.); veselova75@rambler.ru (S.V.); bengal2@yandex.ru (G.B.)

**Keywords:** *Bacillus subtilis*, *Leptinotarsa decemlineata*, *Solanum tuberosum*, abscisic acid, zeatin, zeatin–riboside, indole-3-acetic acid, endophyte

## Abstract

Plant damage caused by defoliating insects has a long-term negative effect on plant growth and productivity. Consequently, the restoration of plant growth after exposure to pathogens or pests is the main indicator of the effectiveness of the implemented defense reactions. A short-term *Leptinotarsa decemlineata* Say attack on potato tube-grown plantlets (*Solanum tuberosum* L.) led to a reduction of both the length and mass of the shoots in 9 days. The decrease of the content of phytohormones—indole-3-acetic acid (IAA), abscisic acid (ABA), zeatin and zeatin–riboside—in shoots of damaged potato plants was found. Endophytic strain *Bacillus subtilis* 26D (Cohn) is capable of secreting up to 83.6 ng/mL IAA and up to 150 ng/mL cytokinins into the culture medium. Inoculation of potato plants with cells of the *B. subtilis* 26D increases zeatin–riboside content in shoots and the mass of roots of undamaged plants, but does not influence content of IAA and ABA and growth of shoots. The presence of *B. subtilis* 26D in plant tissues promoted a rapid recovery of the growth rates of shoots, as well as the wet and dry mass of roots of plants after the pest attack, which we associate with the maintenance of a high level of IAA, ABA and cytokinins in their tissues.

## 1. Introduction

Damage caused by pathogens and insect pests is among the most important factors that reduce the productivity of agricultural plants. The most effective approaches to reduce such negative impacts are the use of pesticides or the development of resistant varieties by either classical breeding or genetic engineering. Damage caused by defoliating insects reduces the photosynthetically active part of the biomass of plants, and thus plant productivity [1], as the plants must utilize additional required energy to overcome this effect. In the majority of plant species, the activation of defense systems leads to interruption of growth—this effect is commonly known as the growth–defense trade-off [2]. While mechanisms of plant defense against pest insects have been extensively investigated, little attention has been paid to the mechanisms of restoring the growth characteristics of plants after damage caused by defoliators, although this is the most important condition for the formation of stable yields.

The Colorado potato beetle (CPB) *Leptinotarsa decemlineata* Say (Coleoptera, Chrysomelidae) is harmful and adaptive to changing environmental factors, a defoliating insect known as an aggressive invader and for its capacity to rapidly develop resistance to insecticides [1]. All modern approaches for control of CPBs including development of Bt-crops, RNAi-based technologies, and applications of *Bacillus thuringiensis,* etc. are based on the idea of pest eradication [3] but none examine the potential for stimulation of plant growth after insect damage.

The multifunctional effect of Plant Growth Promoting Microorganisms (PGPM) is associated with their ability to increase the availability of minerals for plants, to stimulate plant defense mechanisms to pathogens and phytophagous insects, and to have a direct toxic effect on the pest organism [4]. Some PGPM strains are able to produce phytohormones (auxins, cytokinins), that not only regulate plant growth and development, but also participate in the priming of plant defenses [4,5].

Jasmonic acid (JA) and its active form jasmonoyl-L-isoleucine are currently considered key players in the induction of plant defense reactions and regulation of mechanisms of systemic resistance against defoliating insects [6]. However, implication of classical hormones such as ABA, cytokinins and IAA for reparation of plants damaged by insect pests is not clear [2,7]. The importance of these data for the development of modern integrated protocols that take into account the physiological parameters of plants to increase the efficiency and environmental safety of agrobiocenoses certainly is non-contentious.

The strain *B. subtilis* IB-22 produces zeatin–ribosides [8] and can stimulate more than a 10-fold accumulation of cytokinins in lettuce plants inoculated with this strain. Similarly, the auxin-producing strain *B. methylotrophicus* M4-96 isolated from the roots of maize stimulated the growth of *Arabidopsis thaliana* plants inoculated with this strain [9]. The same compound determined the growth-stimulating effect of the strains *B. amyloliquefaciens* FZB42 [10] and *B. amyloliquefaciens* SQR9 [11]. The ability of the bacterial strain *B. aryabhattai* SRB02 to stimulate the growth of *Glycine max* plants was associated with their production of auxins, zeatin, gibberellins, and ABA. *B. aryabhattai* SRB02-treated plants showed heat-stress tolerance and produced consistent levels of ABA under heat stress [12].

A number of strains of the genus *Bacillus* are able to penetrate into the internal tissues of plants, i.e., exist endophytically. Endophytes can provide benefits to the host plant and under diverse environmental conditions they are capable of interacting with plants more efficiently than rhizospheric PGPM [13]. Enrichment of plants with endophytes can be more advantageous than the cultivation of transgenic plants resistant to pathogens [14]. Implementing artificial plant microbiomes with PGPM that are capable of releasing their metabolites immediately in plant tissues can become a challenging alternative to conventional methods of plant protection and growth regulation [15].

In our previous work we showed that the quantity of *B. subtilis* 26D in shoots of potato plants was about 10^5^ cells/mg of fresh weight [15]. Thus, consuming plants containing *B. subtilis* 26D by CPBs resulted in 38.1% larvae death. The presence of endophytic *B. subtilis* 26D strain cells in potato plants enhanced the level of transcripts of jasmonate-biosynthesis genes in plant leaves damaged by *L. decemlineata* [16] and decreased the population density of CPBs under the field conditions [15]. In this context, the question about the influence of *B. subtilis* 26D on growth parameters of potato plants is very important. Is it possible to avoid plant defense/growth trade-offs using endophytic microorganisms?

The aim of this work was to investigate the effect of preliminary inoculation of potato with endophytic bacteria *B. subtilis* 26D and to determine the levels of the phytohormones and growth characteristics of the plants after damage caused by CPBs.

## 2. Results

### 2.1. Level of Phytohormones in the Culture Medium of B. subtilis 26D

It was found that *B. subtilis* 26D is capable of secreting 83.6 ng/mL IAA, 103.8 ng/mL zeatin and 46.2 ng/mL zeatin–riboside into the liquid culture medium (Table 1). ABA was absent in the culture medium of *B. subtilis* 26D. Sterile LB medium did not contain phytohormones (Table 1).

### 2.2. Influence of B. subtilis 26D and CPB on the Growth Characteristics of Potato Plants

Shoot length of undamaged plants treated with bacterial suspension did not exceed the control values. The short-term exposure of water-treated potato plants to CPB feeding caused a significant (Appendix A) decrease of growth of their shoots from the 1st to the 9th day after CPB damage (Figure 1A). Importantly, the presence of endophytic *B. subtilis* 26D in tissues of damaged plants led to a reduced negative effect of the CPB. In the latter case, the statistically significant decrease of shoot length as compared to control plants was observed only on the 9th day after exposure of plants to CPB damage (Appendix A). Two-way ANOVA showed that the length of plant shoots was affected by CPB damage at all time points after damage and their interaction (CPB and *B. subtilis* 26D treatment) just on the 9th day after damage (Appendix A).

In both cases, the additive length of shoots of damaged plants significantly decreased on the 1st day after their injury (Appendix A). However, already on the 3rd day after damage, plants containing endophytic *B. subtilis* 26D showed practically the same growth addition as plants that were not subjected to stress (Figure 1B), while the additional growth of CPB-damaged plants that were untreated with bacterial cells was lower than this parameter of control ones by almost 60%. Later, this parameter of growth of *B. subtilis* 26D-strain treated plants that were exposed to CPB damage decelerated, but the resulting addition of length of shoots of these plants was not significantly different from control scores on the 9th day of the observation. Two-way ANOVA showed that the additional length of plant shoots was affected by CPB damage in the 1st, 3rd and 6th days after damage and interaction of two factors (CPBs and *B. subtilis* 26D treatment) just on the 3rd and 6th days after damage (Appendix A).

As seen from Figure 2A, the weight of shoots of *B. subtilis* 26D-treated undamaged plants did not show significant differences from control plants, but the fresh weights of their roots exceeded the control values by more than 30%, and the dry weights exceeded those values by 15%. Damage caused by CPBs to water-treated plants resulted in a decrease in both the fresh and dry weights of shoots but did not affect the weight of plant roots. Both the fresh and dry weights of shoots of damaged plants containing endophytic *B. subtilis* 26D cells also did not differ from the control ones. The fresh weight of the roots was at the level of undamaged plants of this variant, and the dry weight of roots was slightly, but statistically significantly (*p* ≤ 0.05) higher than the dry weight of roots of undamaged plants treated with *B. subtilis* 26D cells. According to two-way ANOVA, fresh and dry weights of plant shoots were affected just by CPB damage, but fresh and dry weights of roots were affected by *B. subtilis* 26D and interaction of CPBs and *B. subtilis* 26D treatment (Appendix A).

### 2.3. Influence of B. subtilis 26D and CPBs on the Content of Phytohormones in Potato Plants

CPB damage caused a decrease of the level of IAA in potato plants within 3 days after damage (Figure 3A). The level of ABA was reduced on the 1st and 2nd days after exposure to the pest attack, but by the 3rd day after exposure it returned to the control values. In the absence of the stress factor, treatments of plants with *B. subtilis* 26D bacterial cells did not affect the content of IAA and ABA in plants (Appendix A). After contact with CPBs, levels of both IAA and ABA were consistently higher in *B. subtilis* 26D-containing plants than in control ones. Thus, the ABA level in *B. subtilis* 26D-treated plants after CPB damage was higher than in undamaged plants by more than 40% (Figure 3B) on the 2nd and 3rd days after plants were attacked. Two-way ANOVA demonstrated that the level of IAA was affected by CPB damage at all time points after damage. The level of ABA was influenced by CPBs on the 1st and 2nd days. The interaction of two factors (CPBs and *B. subtilis* 26D treatment) influenced both these parameters during 3 days after the damage (Appendix A). *B. subtilis* 26D prevented a decrease of IAA and ABA levels caused by CPB attacks.

It was found that on the 1st and 2nd days the level of zeatin (the active form of cytokinins) decreased in water-treated plants damaged by CPBs, after which its content returned to the control level (Figure 4A). In intact plants, treatment with *B. subtilis* 26D had no effect on zeatin content (about 25 ng/g). On the 1st day post damage caused by CPBs, the level of zeatin in *B. subtilis* 26D-treated plants remained equal to the control means and then it increased to 75 ng/g on the 3rd day post damage.

The level of zeatin–riboside in potato plants that did not contain endophytes was about a third of the control level on the 1st day post damage and then it was about 2/3 of zeatin–riboside content in undamaged plants (Figure 4B). Treatment of plants with *B. subtilis* 26D cells increased the content of zeatin–riboside in intact plants, as evidenced by two-way ANOVA (Appendix A).

On the 1st day post damage caused by CPBs, the content of zeatin–riboside in *B. subtilis* 26D-treated plants remained equal to the control means and then it increased by a factor of 1.5 on the 2nd and the 3rd day after damage. Thus, *B. subtilis* 26D can stimulate levels of cytokinins in plants after CPB attacks.

Using two-way ANOVA we showed that the level of zeatin and zeatin–riboside was affected by CPBs on the 1st and 2nd days after damage; CPB and *B. subtilis* 26D treatment interaction influenced these parameters on the 2nd and 3rd days after damage (Appendix A). Treatment of plants with *B. subtilis* 26D influenced the level of zeatin–riboside in all timepoints (Appendix A).

## 3. Discussion

It was found that even after a short-time of CPB exposure a serious decrease of growth of potato plants was observed. The study showed that treatment of potato plants with cells of the endophytic strain *B. subtilis* 26D promoted the restoration of plant growth after damage caused by CPBs. Previously, data on the alterations of plant growth by bacteria of the genus *Bacillus* under the influence of biotic and abiotic factors were summarized in the reviews [15,17]. However, investigations of the effect of PGPM on productivity of plants that were damaged by insect pests emphasize the direct insecticidal effect of strains under study [18] or their ability to induce plant resistance [16], but not their effect on the growth of damaged plants.

Treatment of tomato plants with the rhizosphere bacteria *Pseudomonas fluorescens* CHA0 prevented the decrease of plant growth, as well as the fresh and dry weight of shoots and roots, induced by nematodes *Meloidogyne javanica* [19]. However, in the latter case the observed prevention could be due to a decrease of the pest reproduction rate on the roots of treated plants. Saleem et al. [20] showed that *Arabidopsis* plants inoculated with *B. cereus* had higher fitness than uninoculated control ones under the influence of fungus gnats (*Bradysia* spp.).

The secretion of growth-stimulating compounds such as auxins and cytokinins by PGPM is well-known [8,11,21]. However, the question of the source of phytohormones in plants containing endophytes in internal tissues remains open. Endophytic bacteria *B. subtilis* 26D produce a lower volume of cytokinins than super-producers of cytokinins, such as rhizospheric strains *B. subtilis* IB-15 (250 ng/mL) and *B. subtilis* IB-22 (600 ng/mL) [21]. This level is apparently sufficient for the manifestation of growth-compensating activity capable of existing endophytically *B. subtilis* 26D. The endophytic strain *B. subtilis* LK14 isolated from the tissues of *S. lycopersicum* plants produced 57.75 ng IAA/mL of culture medium [22], while bacteria isolated from the rhizosphere produced 100–700 ng IAA/mL of culture medium [23]. However, attention is drawn to the absence of differences in the level of IAA and zeatin in untreated plants and plants treated with bacterial cells without CPB damage. Still the level of zeatin–riboside in *B. subtilis* 26D-treated plants was higher than in control ones, and it is not impossible that cells of bacteria under investigation secreted this transport form of cytokinins in plant tissues. Since bacteria used in the present experiments did not secrete ABA in vitro, changes in ABA level in inoculated plants after CPB damage were not due to the uptake of bacterial ABA, but due to the synthesis by plants itself.

Auxins and cytokinins, considered primarily as phytohormones that regulate plant growth and development, are also involved in the formation of defense reactions against various environmental factors, including attacks of pathogens and pests. It was revealed that after imitation of damage caused by *Manduca sexta* caterpillars the rosette diameter and the number of flowers of *Nicotiana attenuata* plants treated with exogenous IAA increased in comparison with damaged plants treated with water [24]. Yan et al. [25] showed that near the surface of the root transition zone of *Arabidopsis*, the net of auxin flux decreased after a *Plutella xylostella* caterpillar attack. Transcript levels of auxin transporter genes PIN1, PIN2, PIN3, PIN7, and AUX1 were also reduced after the exposure of plants to pest eating. It was assumed that the reduction of growth after insect attack was likely associated with a decrease of auxin transport. Instead, the production of IAA by bacteria is mostly responsible for their root-growth-promoting effects. Thus, *Arabidopsis thaliana* inoculation with IAA-producing bacteria *Azospirillum brasilense* Sp245 increased the number of lateral roots and fresh weight of roots under the normal conditions, and the auxin biosynthesis mutant of this strain did not [26].

In this work, we showed that damage caused by CPBs reduced both plant growth and the level of IAA therein; however, treatment with the suspension of *B. subtilis* 26D cells prevented the decrease of these parameters. Under the influence of the investigated strain, the mass of plant roots increased both under normal conditions and after the pest attack. In contrast to IAA, cytokinins are regarded as an inhibitor of root growth, in particular by promoting cell differentiation in the root apical meristem [27]. But it was found that zeatin–ribosides in bacterial cultural media exist as complexes with high molecular weight polysaccharides, which may prevent their sharp inhibitory effect on root growth [8].

The role of cytokinins is also associated with the availability of macro- and micronutrients and nitrogen, which can have a decisive effect on the growth and development of plants and insects feeding on them [28,29]. Treatment of plants with *B. subtilis* 26D prevented the decrease of the level of zeatin, which was observed after damage by the CPBs in water-treated plants, and contributed to the more than 2-fold increase of this parameter on the 3rd day after contact with the pest. At the same time, the content of zeatin–riboside in intact plants containing cells of bacteria increased. Mechanical injury of *N. attenuata* leaves induced transcriptional changes in many genes of biosynthesis and the signaling pathway of cytokinins, and treatment of plants with zeatin–riboside caused an increase in the content of transcripts of jasmonate-dependent genes, including content of transcripts of gene encoding the trypsin inhibitor, that can repress digestive ferments of insects [28].

Dervinis et al. [29] showed that pretreatment of poplar plants with synthetic cytokinins (6-benzylaminopurine) led to an increase in the content of jasmonic acid and the transcriptional activity of enzymes of its biosynthesis after leaves were exposed to mechanical damage, as well as to a decrease of the mass of gypsy moth *Lymantria dispar* caterpillars receiving 6-benzylaminopurine-treated leaves for food. Thus, the role of cytokinins in plant protection from insects causing extensive mechanical damage may be fundamentally different from the well-known effect of cytokinins produced by gall-forming insects [30]. Thus, cytokinins can promote stomatal opening, stimulate shoot growth and decrease root growth [4]. Arkhipova et al. [8] showed that inoculation with cytokinin-producing bacteria *B. subtilis* IB-22 prevented the decrease of cytokinins level in lettuce plants under the influence of soil drying, and stimulated shoot growth and shoot and root weight. The observed growth promoting effect of these cytokinins on stomatal conductance was attributed to increases in shoot ABA content that *B. subtilis* IB-22 also induced.

In rice plants that overexpressed the protein kinase gene *OsMKK3*, the ABA level increased after damage caused by the brown planthopper *Nilaparvata lugens* [31]. In this case, the survival rate of the pest decreased, as did the height of damaged plants; however, the chlorophyll content in attacked plants did not change, as compared with damaged wild-type plants. In our study, the ABA content in the *B. subtilis* 26D-treated plants increased after the CPB attack, but the length of plants remained at the level of undamaged water-treated plants. The restoration of the plant-growth rate, as well as an increase of the mass of roots in plants containing endophytic *B. subtilis* 26D cells, in which the ABA level was increased, can be indirectly associated with overcoming the stress caused by leaf wilting, which occurs under the defoliation caused by insects [32].

According to the literature, ABA plays an important role in the restoration of growth processes after plant exposure to stress factors. For example, an increase of air temperature led to the termination of leaf elongation in both wild-type barley plants and its ABA-deficient mutant (AZ34), but thereafter the growth rate of leaves of wild-type plants was completely restored [33]. In addition, Arkhipova et al. [7] concluded that *B. subtilis* IB-22 bacteria stimulated the ABA content in shoots of wheat plants and contributed to the increase in the mass of their root system, as in the presented study.

The level of ABA in corn plants (*Zea mays*) increased during the attack of Western corn rootworms (*Diabrotica virgifera virgifera*), but not after mechanical injury caused to the roots [29]. In *Arabidopsis,* after mechanically simulated damage by the desert locust (*Schistocerca gregaria*) the ABA content also did not change [34]. Thus, ABA accumulation can be the common process for plant resistance to insect attacks. In the model plant *Arabidopsis*, the signaling pathway of the defense response induced by JA consists of the MYC and ERF branches, which are regulated by ABA and ethylene, respectively. At the same time, the ethylene-dependent ERF branch is associated with plant protection against necrotrophic pathogens, while the MYC branch, which is additionally regulated by ABA, is associated with wounds or pest damages [35,36].

*Arabidopsis* mutant *aba2-1* with a low ABA level demonstrated low expression of the marker of MYC-pathway gene *VSP2* and high expression of the marker gene of ERF-pathway PDF1.2 within 30 h after injury caused by caterpillars *Pieris rapae* [35]. Indeed, in our previous work, the mortality of CPB larvae that were directly treated by suspension of *B. subtilis* 26D bacteria exceeded the control values by no more than 15%, while CPB larvae that ate plants containing the endophytic *B. subtilis* 26D strain exceeded these values by no more than 35% at the same time point after exposure [16], which indicated the effectiveness of induced plant defense reactions.

## 4. Conclusions

We demonstrated the significant decrease of growth of plant shoots that was accompanied by the decrease of IAA, ABA and cytokinins level and, consequently, the importance of the ability of *B. subtilis* 26D to maintain phytohormones levels in plants and to promote plants’ adaptation to CPB-caused damage. The ability of bacteria to increase phytohormones content in potato shoots after the damage caused by defoliating phytophage under investigation was shown for the first time, and this response can promote both the root growth and reparation of shoots growth. We detected that the effect of *B. subtilis* 26D on shoots parameters and the levels of IAA, ABA and zeatin was significant only when plants were affected by CPBs, and therefore the impact of this phytohormone-producing strain cannot deform the phytohormonal status of plants under the normal conditions. We showed that treatment of plants with *B. subtilis* 26D led to an increase of the level of zeatin–riboside in damaged and nondamaged plants, which could be the reason for the promotion of rooting system development. Identification of mechanisms providing a variety of bacterial effects on reparation of plant growth after insect pests’ damage and after the influence of other factors should be the subject of further research. However, our results indicate the importance of bacterial regulation of the hormonal status of plants for the activation of plant growth after defoliating insects attacks, as well as the significant contribution of zeatin–riboside accumulation in shoots for the promoting effect of *B. subtilis* 26 D on the rooting system of plants.

## 5. Materials and Methods

### 5.1. Plant, Microbe and Insect Material

Plants: sterile *Solanum tuberosum* L. plants cultivar Early Rose obtained by microcloning technology and grown in tubes with Murashige and Skoog medium in a KBW E6 climatic chamber (Binder GmbH, Tuttlingen, Germany) with a 16 h light period at 20–22 °C for 21 days were used.

Bacteria: Gram-positive aerobic *B. subtilis* 26D strain from the collection of the Laboratory of Biochemistry of Plant Immunity of the Institute of Biochemistry and Genetics UFRC [37] were used. Bacteria were grown on liquid Luria-Bertani (LB) medium (1% tryptone, 0.5% yeast extract, 0.5% NaCl) at 20–22 °C using laboratory shakers (120 rpm).

System “Plant + endophyte”: 14-day-old plants were inoculated with 5 μL of *B. subtilis* 26D suspension on the stem neighbor of the zone of formation of adventitious roots according to the previously described method [16]. Concentration of bacterial cells was 10^8^ cells/mL. Plants were grown in a gnotobiotic system for 7 days, and then the number of endophytic bacteria stood at 10^5^ cells/mg of fresh weight. Part of the plants was treated with 5 μL of distilled water (distiller A1110, LLC Liston, Zhukov, Russia) and was used as controls in all experiments.

*L. decemlineata*: adults were taken from seed potato plants grown on a pesticide-free field of the Chishminsky Plant Breeding Center of the Bashkir Research Institute of Agriculture of UFRC RAS in 2020 (Republic of Bashkortostan, Chishmi, 54°34′49.6″ N 55°25′35.5″ E). Insects were kept under laboratory conditions in pairs in petri dishes. Distilled water in plastic tubes (“Eppendorf”, Hamburg, Germany) and fresh potato leaves were replaced as needed.

### 5.2. Plant Damaging

In order to damage plants treated with water or *B. subtilis* 26D suspension, adult CPBs were placed in test tubes with cultivated plants for 3–5 min, one at a time [16]. During this time, one adult ate about 2–3 mg of leaves of one plant, after which the insect was removed. The level of tissue damage was controlled visually (2–3 mg is the weight of 1/2 leaf) and using CE224-C analytical weight scales (Sartogosm, St. Petersburg, Russia). Plants treated with water and *B. subtilis* 26D, which were not exposed to CPB damage, were used for investigation of all parameters in intact plants. Water-treated intact plants were used as controls in all experiments.

### 5.3. Plant Growth Parameters

The length of shoots of each plantlet from the base of the stem to the apical bud was measured before the contact with the pest (0 point on graphs), 1, 3, 6 and 9 days after injury without being removed from the test tube (using aseptic technique). The size of the increment of shoot length, which was grown over consecutive time spans, was calculated for each plant. On the 9th day after exposure to CPBs, damaged and intact plants were removed from test tubes; their roots were washed from the agar medium by washing with running water, and then thoroughly dried on filter paper. Roots and shoots were weighed separately on CE224-C analytical weigh scales (Sartogosm, St. Petersburg, Russia), then the weighed portions were dried twice at 60 °C for 1 h in the SNOL 67/350 dry heat oven (AB Umega, Utena, Lithuania) to measure the dry weight.

### 5.4. Phytohormones Assay

Bacterial and plant materials processing. Liquid culture medium obtainable by cultivating *B. subtilis* 26D was collected at the late logarithmic growth phase or at the beginning of the stationary phase (on the third day) and centrifuged at 4000× *g* for 20 min in Avanti J-E centrifuge (Beckman Coulter, Bray, OK, USA). The supernatant was analyzed for the content of phytohormones (cytokinins and IAA).

To analyze the levels of cytokinins, IAA, and ABA in shoots (stem with leaves) of test-tube potato plants on the 1st, 2nd and 3rd day after exposure to CPBs, the plant material was homogenized in 80% ethanol (wet weight:volume of ethanol = 1:10) and extracted for 16–20 h at 4 °C. Then the extract was separated by centrifugation at 4000× *g* for 20 min in an Avanti JE centrifuge (Beckman Coulter, Bray, OK, USA) and evaporated to an aqueous residue [37,38].

Cytokinins assay. Cytokinins from 2 mL of the supernatant of the bacterial culture liquid were twice extracted with n-butyl alcohol in a 2:1 ratio (aqueous phase/organic phase) [37]. The extract was evaporated to dryness. Cytokinins from the aqueous residue of plant extract were concentrated by passing through a pre-wetted C18 reverse phase column (Bond-Elut, RP-C18; Varian Ltd., Walton-on-Thames, UK), eluates were evaporated to dryness.

After solvent evaporation, the dry residues in either case were dissolved in 0.02 mL of 80% ethanol and applied to silica gel thin-layer chromatography plates 60 F-254 (Merck, Darmstadt, Germany). Chromatography was carried out in the solvent system butanol:ammonia:water (6:1:2) to separate and assay zeatin–riboside (Rf 0.4–0.5) and zeatin (Rf 0.6–0.7). Zones that contained cytokinins (based on position of standards) were eluted with 0.1 M pH 7.4 phosphate buffer for 16 h, then the silica gel was removed by centrifugation at 10,000× *g* for 10 min in 5415K centrifuge (Eppendorf, Hamburg, Germany). Aliquots of supernatant were added to microplate wells in serial dilutions to assay the cytokinins level [39]. Specificity of antibodies against zeatin–riboside used for ELISA was described previously [40]. The reliability of the hormone immunoassay was confirmed using the dilution test, and results correlated with the results of high performance liquid chromatography (HPLC) in combination with mass spectrometry [8,40].

IAA and ABA assays. IAA and ABA were extracted from aqueous residues (which were obtained after bacterial and plant materials processing) with diethyl ether as described previously [14]. ABA and IAA were separated with diethyl ether from the aqueous residue, after their dilution with distilled water and acidification with HCl to pH 2.5 (organic phase:aqueous phase being/3:1). Then, hormones were transferred from the organic phase into a solution of NaHCO_3_ (organic phase: aqueous phase being/1:3), and were re-extracted from the acidified aqueous phase with diethyl ether. At the next stage, the samples were methylated using diazomethane and evaporated to dryness. Dried residues were dissolved in 80% ethanol. An IAA and ABA quantitative assay was performed with ELISA using specific antibodies as described previously [33]. Phytohormones content in bacterial culture medium and plant shoots was calculated per ml culture medium and mg of wet weight, respectively. The sufficiency of hormones purification prior to immunoassay was proved by studying the chromatographic distribution of the immunoreactive material previously [8].

### 5.5. Statistical Analysis

A total of 30 plants were used in each variant for growth-parameter estimation. For estimation of phytohormones content, six replicates were used in each variant and shoots of four plants were sampled for each repetition. Phytohormone content in the culture medium of *B. subtilis* 26D was investigated in 10 independent flasks.

In order to assess the effect of the treatments on growth parameters and phytohormones levels, 2-way ANOVA was used, setting *B. subtilis* 26D treatment and CPB damage as fixed factors (See Appendix A).

Data showed mean values with standard errors (±SE). Asterisks indicate significant differences among treatments and intact water-treated plants in the same day according to Tukey’s HSD multiple range tests at *p* ≤ 0.05. Statistica 12.0 (Stat Soft, Tulsa, OK, USA) and Excel 2010 (Microsoft, Redmond, WA, USA) software were used.

## Figures and Tables

**Figure 1 plants-10-00923-f001:**
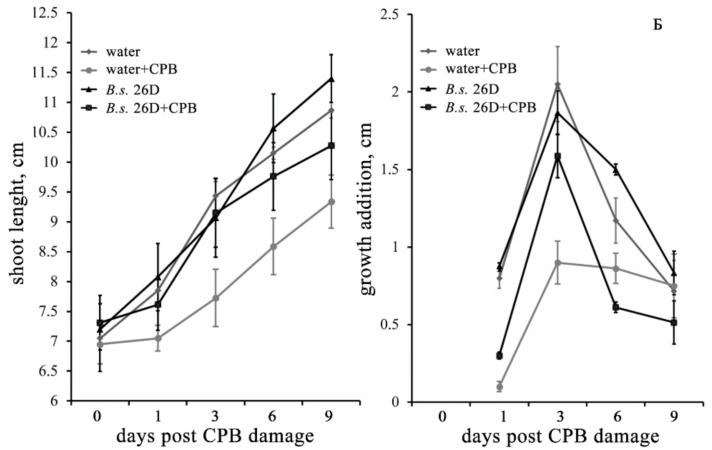
Influence of damage caused by CPB on the length of shoots (**A**) and the growth addition of shoots (**B**) of potato plants treated with water or suspension of *B. subtilis* 26D. The values are means, and the vertical bars represent standard errors. Data analyzed using two-way ANOVA with Tukey’s post hoc test. Asterisks indicate means statistically different from the control at *p* ≤ 0.05. CPB—plants damaged by Colorado potato beetle.

**Figure 2 plants-10-00923-f002:**
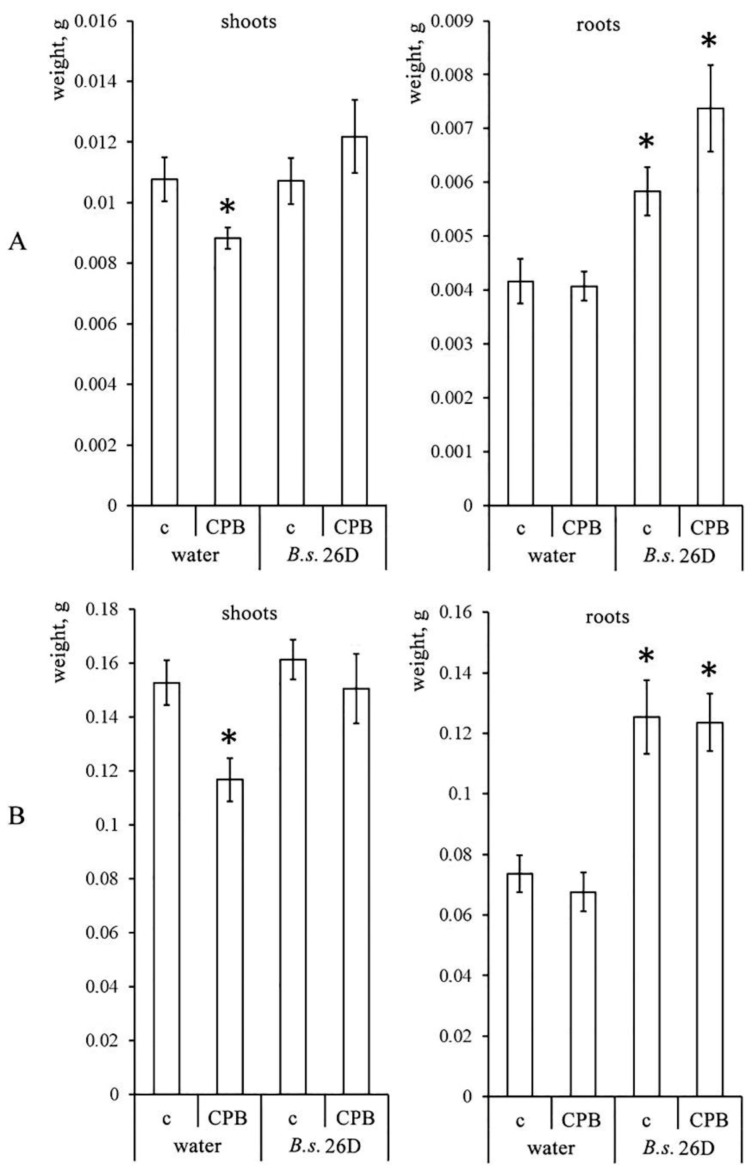
Influence of *B. subtilis* 26D on fresh (**A**) and dry (**B**) weights of shoots and roots of potato plants on the 9th day post damage caused by CPBs. The values are means, and the vertical bars represent standard errors. Data analyzed using two-way ANOVA with Tukey’s post hoc test. Asterisks indicate means statistically different from the control at *p* ≤ 0.05. c—control plants (water-treated intact plants); CPB—plants damaged by Colorado potato beetle.

**Figure 3 plants-10-00923-f003:**
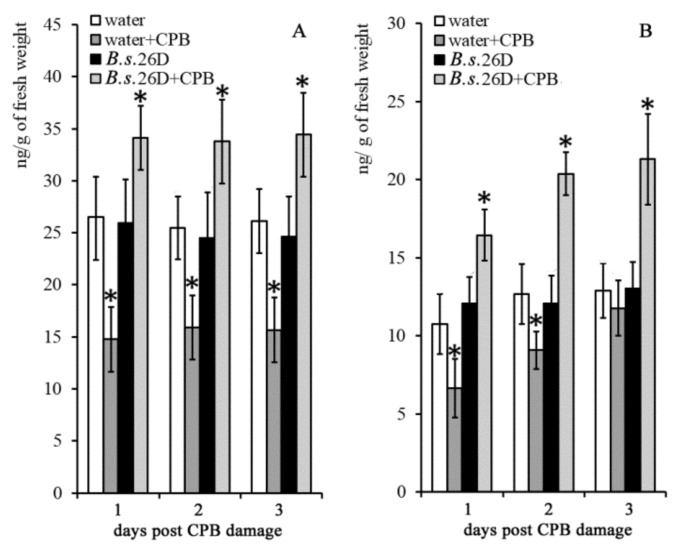
Influence of *B. subtilis* 26D on the level of IAA (**A**) and ABA (**B**) in potato plants on the 1st, 2nd and 3rd day after damage caused by CPBs. The values are means, and the vertical bars represent standard errors. Data analyzed using two-way ANOVA with Tukey’s post hoc test. Asterisks indicate means statistically different from the control at *p* ≤ 0.05. c—control plants (water-treated intact plants); CPB—plants damaged by Colorado potato beetle.

**Figure 4 plants-10-00923-f004:**
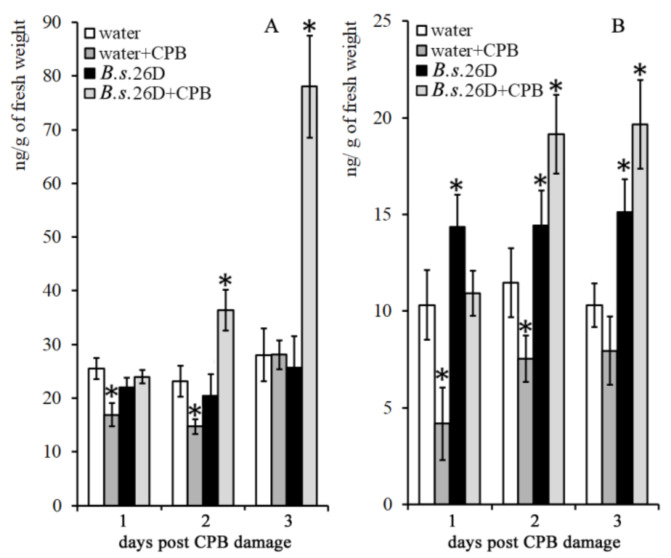
Influence of *B. subtilis* 26D on the level of zeatin (**A**) and zeatin–riboside (**B**) in potato plants on the 1st, 2nd and 3rd day after damage caused by CPBs. The values are means, and the vertical bars represent standard errors. Data analyzed using two-way ANOVA with Tukey’s post hoc test. Asterisks indicate means statistically different from the control at *p* ≤ 0.05. c—control plants (water-treated intact plants); CPB—plants damaged by Colorado potato beetle.

**Table 1 plants-10-00923-t001:** Content of phytohormones in the culture medium of *B. subtilis* 26D strain.

	Phytohormones Level, ng/mL of Culture Medium			
	IAA	ABA	Zeatin	Zeatin-Riboside
*B. subtilis* 26D	83.6 ± 3.9	0	103.8 ± 12.9	46.2 ± 6.0
Sterile LB	0	0	0	0

## Data Availability

Data available in a publicly accessible repository.

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
