# Peer review of "Endophytic Strain Bacillus subtilis 26D Increases Levels of Phytohormones and Repairs Growth of Potato Plants after Colorado Potato Beetle Damage"

_plants, 2021, doi:10.3390/plants10050923_

Round 1

Reviewer 1 Report

The manuscript entitled: Endophytic strain Bacillus subtilis 26D increases levels of phytohormones and repairs growth of potato plants after Colorado potato beetle damage” presents the results of studies which, in my opinion, are interesting and contemporary.

The manuscript does not describe many parameters but contains important information regarding the contribution of hormones and growth regulators to insect attack. The subject of the studies can be included in the current research trends in biocontrol.

Studies were designed and described well. The introduction is adequate to the subject of the manuscript. The description of the methods and results is clear. The discussion is written well.

Therefore, I recommend the manuscript to be published in Plants after minor revision.

Below I present only several suggestions.

Line 154: Is „promotes”, should be „promoted”

Line 205: Is „and cytokinins”, should be „, cytokinins”

Line 207: Is „zeatin-riboside”, should be „zeatin-ribosides”. Is „complex”, should be, „complexes”

Line 216: Is „containing endophytic cells”, should be „cells of endophytic bacteria”

Line 219: Is „causes”, should be „caused”

Line 220: Is „repress”, should be „repressed”

Line 235: Is „overexpress”, should be „overexpressed”

Line 249-251: Please, rewrite this sentence to be more clear.

Line 256: „only” should be removed. The sentence suggests that ABA content can not increase in plants stressed with other biotic and abiotic factors.

Line 287: „was” should be removed

Line 356: Add comma before „respectively”

In the section „Statistical analysis” the information about the number of replicates might be added (not obligatory)

Author Response

Point 1. Line 154: Is „promotes”, should be „promoted”

Line 205: Is „and cytokinins”, should be „, cytokinins”

Line 207: Is „zeatin-riboside”, should be „zeatin-ribosides”. Is „complex”, should be, „complexes”

Line 216: Is „containing endophytic cells”, should be „cells of endophytic bacteria”

Line 219: Is „causes”, should be „caused”

Line 220: Is „repress”, should be „repressed”

Line 235: Is „overexpress”, should be „overexpressed”

Line 249-251: Please, rewrite this sentence to be more clear.

Line 256: „only” should be removed. The sentence suggests that ABA content can not increase in plants stressed with other biotic and abiotic factors.

Line 287: „was” should be removed

Line 356: Add comma before „respectively”

Response 1: We are very grateful to the reviewer for attention to our manuscript and important remarks. We are sorry for grammatical errors.

Point 2. In the section „Statistical analysis” the information about the number of replicates might be added (not obligatory)

Response 2: The information about the number of replicates was included in special paragraphs (about plant growth measuring, phytohormones content measuring etc.). We would prefer not to post the same information In „Statistical analysis”, or move it (since it can obstruct understanding of each method).

Reviewer 2 Report

Dear Authors,
The following statements are prepared in accordance with the expectations of the review form.
At the outset, it is worth emphasizing that the manuscript contribute new knowledge. I would like to make a general comment, the inclusion of which will affect the layout and content of the manuscript. My comments are not only suggestions.
The introduction provide sufficient background for the main purpose of the manuscript. Please consider combining the 3rd and 5th paragraphs.
The study design is appropriate, but I disagree with the analysis and presentation of the data. According to the manuscript, a one-way analysis of variance was performed, but no information about the levels of the factors was provided. As I assume the plat treatment with four levels was a factor. In my opinion, a two-way ANOVA with interaction of factors should have been performed. The first factor should have been "plant treatment" and the second factor should have been "dpd – days post damage". This approach would have allowed on the independent evaluation of the influence of "plant treatment " and "dpd" on the analyzed features of the plants. Such an evaluation is missing in the present work! These influences should be presented in tabular form as homogeneous groups independently for "plant treatment" and "dpd". A two-way ANOVA would have allowed on the analysis of the interaction between "plant treatment" and "dpd". Reader needs the statistical confirmation of the effect of these sources of variation on the analyzed features of the plants. Figure 1 is the result of five one-way ANOVA analyses, but actually presents an interaction of factors (plant treatment x dpd). In my opinion, two-way ANOVA analyses (separately for the indices/features) should be performed. I return again to the question posed above - where are the averages showing clear effects of the main factors ("plant treatment" and "dpd")? The figures showing interaction should be rethought and rebuilt after recalculation the data. Figure 2., how it should be read? Which means “on the same time point” should be compare? The data is only for the ninth days post damage. How the plant treatment influence a weight? A similar approach and similar errors occurred in figure 3. Additionally, if the statement (values labelled by the same character are not significantly different from each other (on the same time point)) is true then the description of the figure is incorrect. The title of figure 4 is probably part of the text. How to read the figure, similarly to figure 3? According to what I have mentioned above, the description of the figure 4 is probably incorrect too. Implementing my suggestions might increase the number of the tables, but at the same time, it would make easier the analyze of the influence of the main factors on the studied traits, as this was done too superficially in the article. The authors revealed some problems with the interaction analyses. What I mentioned above raises my doubts about the correctness of the description of the statistical analyses. The data should be recalculated and the text should be rewritten.
The methods are generally described sufficiently. If my suggestions, mentioned above, are implemented, the chapter "Statistical Analysis" must be rebuilt.
The results are not clearly presented. The "Results" chapter was written according to a one-way ANOVA analysis, which, as I showed earlier, is not the right way to do it. The chapter contains four figures showing the interaction of factors, but none of them (the figures) provide statistical evidence for interaction.
In present form the conclusions (one sentence) are absolutely to short.
A less important remarks was submitted in the manuscript (comments in the margin).
According to what I have mentioned above, my recommendation for the Editor cannot be positive. In this form, the manuscript is not acceptable for publication. If the Editor orders reconsidering the manuscript after major revision, I will not oppose. I suggest resubmitting the manuscript after taking into account the comments.
Yours sincerely,
Reviewer

Author Response

Dear Reviewer,

We are very grateful to you for your attention to our manuscript and important remarks. We regret that we didn't pay a lot of attention to presentation of statistical data. Please find our detailed replies below.

Point 1. The introduction provide sufficient background for the main purpose of the manuscript. Please consider combining the 3rd and 5th paragraphs.

Response 1. The 3rd paragraph was shorten and combined with the 5th paragraph.

Point 2. The study design is appropriate, but I disagree with the analysis and presentation of the data. According to the manuscript, a one-way analysis of variance was performed, but no information about the levels of the factors was provided. As I assume the plat treatment with four levels was a factor. In my opinion, a two-way ANOVA with interaction of factors should have been performed. The first factor should have been "plant treatment" and the second factor should have been "dpd – days post damage". This approach would have allowed on the independent evaluation of the influence of "plant treatment " and "dpd" on the analyzed features of the plants. Such an evaluation is missing in the present work! These influences should be presented in tabular form as homogeneous groups independently for "plant treatment" and "dpd". A two-way ANOVA would have allowed on the analysis of the interaction between "plant treatment" and "dpd". Reader needs the statistical confirmation of the effect of these sources of variation on the analyzed features of the plants. Figure 1 is the result of five one-way ANOVA analyses, but actually presents an interaction of factors (plant treatment x dpd). In my opinion, two-way ANOVA analyses (separately for the indices/features) should be performed. I return again to the question posed above - where are the averages showing clear effects of the main factors ("plant treatment" and "dpd")? The figures showing interaction should be rethought and rebuilt after recalculation the data.

Response 2. In the manuscript we described the influence of two factors the influence of damage caused by Colorado potato beetle (factor 1) on growth and hormones level of plants, treated or non-treated with Bacillus subtilis 26D (factor 2). Consequently, In order to assess the effect of the treatments on growth parameters and phytohormones levels, 2-way ANOVA was used, setting B. subtilis 26D treatment and CPB damage as fixed factors. Circumstantial statistical data was performed in Supplementary, relevant explanations were placed in the text of the article.

Thus, dpd (days post CPB damage) designate only the time point. We regret that that this way of data presentation was unclear. After careful consideration we have decided to give up dpd abbreviation because it could confuse the reader. Dpd was replaced by the designation of days on X-axis in figures (if relevant).

Point 3. Figure 2., how it should be read? Which means “on the same time point” should be compare? The data is only for the ninth days post damage. How the plant treatment influence a weight?

Response 3. We are sorry for inaccuracy in the legend. Weight was investigated on the 9th day after exposure to CPB. It was designated in the legend. The figure was rebuilt (weight of shoots and roots in the first version was showed on combined charts, now it placed in separate charts for shoots and roots).

Point 4. A similar approach and similar errors occurred in figure 3. Additionally, if the statement (values labelled by the same character are not significantly different from each other (on the same time point)) is true then the description of the figure is incorrect.

Response 4. Figures 3 and 4 were rebuilt. Columns and rows were reversed. Dpd was replaced by the designation of days on X-axis. Letters were replaced on asterisks, indicating significant differences among treatments and intact water-treated plants in the same day (control) at p ≤ 0.05. Indeed, data was analyzed on the same time point separately (see Supplementary table 3).

Point 5. The title of figure 4 is probably part of the text. How to read the figure, similarly to figure 3? According to what I have mentioned above, the description of the figure 4 is probably incorrect too.

Response 5. We have confusion about the absence of figure 4 legend and can only assume the technical problem. The legend is placed below just in case:

Figure 4. Influence of B. subtilis 26D on the level of zeatin (A) and zeatin-riboside (B) in potato plants on the 1st, 2nd, and 3rd day after damage caused by CPB. The values are means, and the vertical bars represent standard errors. Data analyzed using two-way ANOVA with Tukey’s post hoc test. Asterisks indicate significant differences among treatments and intact water-treated plants in the same day at  p ≤ 0.05. c – control plants (water-treated intact plants); CPB – plants damaged by Colorado potato beetle.

Figure 4 was rebuild as well as figure 3. Statistical analysis was performed in the same way (Supplementary Table 3)

Point 6. Implementing my suggestions might increase the number of the tables, but at the same time, it would make easier the analyze of the influence of the main factors on the studied traits, as this was done too superficially in the article. The authors revealed some problems with the interaction analyses. What I mentioned above raises my doubts about the correctness of the description of the statistical analyses. The data should be recalculated and the text should be rewritten.

Response 6. Statistical analysis was implemented for all parameters and all time points. Statistical parameters were placed in Supplementary. We regret not having done it earlier. Our data and conclusions did not change, but their presentation was strictly redesigned.

Point 7. The methods are generally described sufficiently. If my suggestions, mentioned above, are implemented, the chapter "Statistical Analysis" must be rebuilt.

Response 7. Statistical Analysis was rebuilt. Data on the number of samples was removed from another paragraphs and placed in Statistical analysis. The ANOVA method description was complemented.

Please find whole paragraph below:

30 plants were used in each variant for growth parameters estimation. For estimation of phytohormones content six replicates were used in each variant, shoots of four plants were sampled for each repetition. Phytohormone content in the culture medium of B. subtilis 26D was investigated in 10 independent flasks.

In order to assess the effect of the treatments on growth parameters and phytohormones levels, 2-way ANOVA was used, setting B. subtilis 26D treatment and CPB damage as fixed factors (See Supplementary).

Data showed mean values with standard errors (±SE). Asterisks indicate significant differences among treatments and intact water-treated plants in the same day according to Tukey's HSD multiple range tests at p ≤ 0.05. Statistica 12.0 (Stat Soft, Russia) and Excel 2010 (Microsoft, USA) software were used.

Point 8. The results are not clearly presented. The "Results" chapter was written according to a oneway ANOVA analysis, which, as I showed earlier, is not the right way to do it. The chapter contains four figures showing the interaction of factors, but none of them (the figures) provide statistical evidence for interaction.

Response 8. Statistical evidences of interaction of two factors (B. subtilis treatment and CPB damage) were presented in Supplementsry.

Point 9. In present form the conclusions (one sentence) are absolutely to short. A less important remarks was submitted in the manuscript (comments in the margin).

Response 9. Conclusions were substantially complemented and separated from the Discussion.

Please find whole paragraph below:

  1. Conclusion

Thus, we demonstrated the the sygnificant decrease of grown of plant shoots, that accompanied by the decrease of IAA, ABA and citokinins level and, consequently, importance of the ability of B. subtilis 26D to maintenance their level for plant adaptation to CPB-caused damage. The ability of bacteria to increase phytohormones content in potato shoots after the damage caused by defoliating phytophage under investigation was shown for the first time, and this response can promote both the root growth and reparation of shoots growth. We detected that the effect of B. subtils 26D on shoots parameters and the levels of IAA, ABA and zeatin was significant only then plants were affected by CPB and therefore the impact of this phytohormone-producing strain can not deform phytohormonal status of plants under the normal conditions. We showed that treatment of plants with B. subtilis 26D led to increase of the level of zeatin-riboside in damaged and non-damaged plants, that can be the reason for the promotion of rooting system development. Identification of mechanisms providing a variety of bacterial effects on reparation of plant growth after insect pests damage and after the influence of another factors should be the subject of further research. However, our results indicate the importance of bacterial regulation of hormonal status of plants for the activation of plant growth after defoliating insects attacks, as well as the significant contribution of zeatin-riboside accumulation in shoots for the rooting system promoting effect of B. subtilis 26 D.

We found some comments in the text. Mistakes were fixed. Some difficult questions about figures and statistical analysis were answered above. Table 1 contains the row “Sterile LB” to avoid some doubts, that phytohormones can presence in the culture medium or its components can disturb the data. It was described in the text (“Sterile LB medium did not contain phytohormones (Table 1)”).

Sincerely yours,

Igor.

Reviewer 3 Report

The manuscript by Sorokan et al. detailing the “Endophytic strain Bacillus subtilis 26D increases levels of phytohormones and repairs growth of potato plants after Colorado potato beetle damage” is overall an interesting study. It examines the recovery of growth of tissue culture grown potato plantlets that were inoculated with a Bacillus species and exposed to feeding stress by Colorado potato beetles. The appropriate controls are included for testing of how the combination of bacteria and feeding stress impacts plant growth recovery.  The analytical methods are appropriate, the number of replications are adequate for comparisons and the data are well analyzed. The result indicate that endophytic bacteria enhance the recovery of the plant by altering several key plant hormones, most specifically increases zeatin-riboside content. The capacity of the bacteria to produce key hormones in culture was demonstrated but it is not clear if these may be contributing to the plant’s response to the stress. The English used within the manuscript can be improved and I have made some suggested changes in the Word version of the manuscript. I am sure that the technical editor can help with this even beyond my attempts. The fact that the bacteria can harm the Colorado potato beetle, and may even kill them, is most interesting.

There are some areas that need to be elucidated:

1)  where are the bacteria – are they on the surface of the tissue culture grown plants or are they really endophytes? Can they provide  evidence that the bacteria are colonizing only the internal compartments of the plants.

2) How was the level of tissue damage controlled to be identical for the treated and control plants?

While this study is interesting and publishable within the context of being a model system, the results would have a much greater impact if the authors took it to the next step and showed that the responses observed also persist within mature potato plants. Tissue culture grown plants are to a large extent in an embryonic state and their metabolism is not comparable to plants grown exposed to natural conditions.

Author Response

Thank you ever so much for taking the time to consider our paper.

Point 1. where are the bacteria – are they on the surface of the tissue culture grown plants or are they really endophytes? Can they provide  evidence that the bacteria are colonizing only the internal compartments of the plants.

Response 1. Strain Bacillus subtilis 26D is facultative mutualist of plants. We previously  reported its ability to exist in internal plant tissues in our previous works (for example, Sorokan, A., Cherepanova, E., Burkhanova, G. et al Endophytic Bacillus spp. as a prospective biological tool for control of viral diseases and non-vector Leptinotarsa decemlineata Say. in Solanum tuberosum L. Front. Microbiol. 2020, 11, 569457. https://doi.org/10.3389/fmicb.2020.569457). Our method is based on counting the CFU of bacteria in homogenates of surface-sterilized plants (when bacteria on the surfaces were eliminated). The description of this protocol you can see in the article above.

In the paper we showed published data on this topic (Thus, in our previous work it was found that the quantity of B. subtilis 26D in shoots of potato plants was about 105 cells / mg of fresh weight [15])

We cannot exclude, that a part of B. subtilis 26D can exist on plant surface, but it’s no doubt, that its cells can lie into plant tissues. In our opinion, obligate symbionts could not be successfully used as biocontrol agents, since their cultivation may be difficult or impossible on a commercial scale. 

Point 2. How was the level of tissue damage controlled to be identical for the treated and control plants?

Response 2: We put some clarity into the section 4.2. “Plant damaging”:

“The level of tissue damage was controlled visually (2-3 mg is the weight of 1/2 leaf) and using CE224-C analytical weight scales (Sartogosm, Russia).”

Round 2

Reviewer 2 Report

Dear Authors,

The manuscript has been noticeably improved. I accept its in present form.

Your sincerely,

Reviewer

Author Response

The manuscript has been verified.